# Lymphocyte activation after a high-intensity street dance class

**Leandro Borges** [1]\*, **Renata Gorjão**[1], **Stuart R. Gray**[2], **Thaís Reis Martins**[1], **Vinicius Coneglian Santos**[1], **Cesar Miguel Momesso** [1], **Tania Cristina Pithon-Curi**[1], **Elaine Hatanaka**[1]

**1** Institute of Physical Activity and Sport Sciences (ICAFE), Cruzeiro do Sul University, São Paulo, SP, Brazil,
**2** Institute of Cardiovascular & Medical Sciences, University of Glasgow, Glasgow, United Kingdom

\* sbleandro@yahoo.com.br

## Abstract

Intense dance training leads to inflammation, which may impair the health and performance of the practitioners. Herein, we evaluate the effect of a single street dancing class on the profile of muscle enzymes, lymphocyte activation, and cell surface CD62L expression. We also investigated the correlation between muscle enzymes, adhesion molecules, and lymphocyte activation in dancers. Fifteen male participants (mean ± standard error: age 22.4 ± 1.08 years, body mass index 24.8 ± 0.69 kg/m$^2$, body fat 12.3 ± 1.52%), who were amateur dancers, had blood samples collected previously and subsequent to a high-intensity street dance class. After the class, dancers showed an increase in total lymphocyte count (2.0-fold), creatine kinase (CK)-NAC (4.87%), and CK-MB (3.36%). We also observed a decrease (2.5-fold) in reactive oxygen species (ROS) produced by lymphocytes, under phorbol myristate acetate-stimulated environments. Following the dance class, CD62L expression in lymphocytes decreased (51.42%), while there was a negative correlation between the intensity of the exercise and CD62L expression (r = -0.73; p = 0.01). Lymphocytes were less responsive to stimuli after a single bout of street dancing, indicating transient immunosuppression.

## Introduction

Street dance involves explosive movements, jumps, accelerations, and decelerations, resulting in acute fatigue and decreased performance in dancers [1]. High-intensity street dance exercise also results in neutrophil dysfunction, accompanied by increased pro-inflammatory cytokines [2]. Moreover, exercise, in general, induces an elevation in total blood counts and lymphocyte proliferative responses [3] and inflammation has been implicated in the genesis of inflammatory joint diseases in dancers [4].

In humans, one approach used to explore lymphocyte distribution is the assessment of cell surface adhesion molecule expression. Such analyses can reveal the trafficking patterns of cells mobilized into the bloodstream during physical exercise [5, 6]. One of these adhesion molecules, sL-selectin (CD62-L) is produced by the shedding of L-selectin from the membrane and sL-selectin acts as an antagonist to the membrane-bound form, preventing the binding of

**Funding:** This research was supported by the Fundação de Amparo à Pesquisa do Estado de São

Paulo (FAPESP) under Grant [2011/21441-0 and 2014/21185-1] (EH); Conselho Nacional de Desenvolvimento Científico e Tecnológico (CNPq) under Grant [306041/2011-1 and 308700/2017-1] (EH); and the Coordenação de Aperfeiçoamento de Pessoal de Nível Superior (CAPES) under Grant [88882.314890/2013-01] (LB). The funders had no role in study design, data collection and analysis, decision to publish, or preparation of the manuscript.

**Competing interests:** The authors have declared that no competing interests exist.

lymphocytes. Although sL-selectin is considered an acute phase marker, chronic inflammation leads to decreased levels of sL-selectin, decreasing immune function [7]. Therefore, sL-selectin and CD62L adhesion molecules are critical components of the immune system which have a role in host defense and chronic injuries.

Herein, we hypothesized that a single session of high-intensity street dance exercise could activate lymphocytes and alter cell surface adhesion molecule expression and that these changes could correlate with muscle injury enzymes and exercise intensity. In the present study, we evaluate the effect of street dancing on lymphocyte counts, CD62L expression in lymphocytes, and reactive oxygen species (ROS) release. We also measured the plasma activity of creatine kinase (CK) and lactate dehydrogenase (LDH) before and after the class.

## Materials and methods

### Participants

Male volunteers (15 in total) signed written informed consent to be included in the research. Experimental procedures were conducted according to the Declaration of Helsinki after approval from the Ethical Committee of the Cruzeiro do Sul University (Certificate Number: 0522013). The volunteers were characterized by (mean ± standard error of the mean (SEM)): weight 70.8± 1.93 kg, height 1.69± 0.02 m, age 22.4± 1.08 years, body mass index (BMI) 24.8± 0.69, body fat 12.3± 1.52% (measured by the tetrapolar bioimpedance device: Biodynamics Corporation, 310, EUA), flexibility measured by sit-and-reach test 37± 2.22 cm, red blood cells $5.4 \pm 0.14$ mil/mm$^3$, white blood cells $9 \pm 0.53$ mil/mm$^3$, hemoglobin $14.7 \pm 0.29$ g/dL, hematocrit $44.2 \pm 0.88$%, mean red cell volume $83 \pm 1.86$ fL and hemoglobin concentration per red blood cell $33.3 \pm 2.40$ percent (blood count corresponds to the resting values). The participants had a dance training/performance history of 3.7± 0.39 hours/day, four days per week, and an average sports experience of 7 years.

Heart rate (HR) was monitored continuously by *Polar FT7M* HR monitors during exercise. For calculation of HR maximum, we used the formula: HR maximum = 220 –age. All participants danced at high intensity for 60 minutes, according to the following intensities based on HR reserve: at 85% in the first 20 minutes of class; at 88.9% at 20–40 minutes of the class and at 89.2% at 40–60 minutes of the class [2]. The training intensity was classified as intense according to the guidelines of the American College of Sports Medicine [8]. In the current research, the dancers did not perform any exercise for 72 hours before the dance class and the later correlations of the study were obtained from the HR reserve of each participant. Participants who were taking medication or had a history of metabolic or immunological diseases were excluded from the study.

### Sample collection and isolation of blood lymphocytes

Before and immediately after class, there was a sample collection of 20 milliliters of venous blood from the antecubital vein. Blood samples were drawn into BD Vacutainer® tubes, containing heparin. All samples were assessed on a single day and the blood collection delay following exercise between participants was approximately 3–4 minutes. As reported previously [9], lymphocytes were isolated from peripheral blood. Phosphate buffer saline (PBS, pH 7.4) was used to dilute the blood samples (1:1), and the diluted suspension layered onto Histopaque-1077 (Sigma Chemical Co., St. Louis, MO, EUA) and centrifuged (at room temperature for 30 min at 400 x *g*). Peripheral blood mononuclear cells (PBMC) were obtained from the interphase. The residual erythrocytes were lysed in a solution containing 150 mM NH4Cl, 10 mM NaHCO3, and 0.1 mM EDTA, pH 7.4. PBS was used to wash the cells. PBMC were preserved in sterile tissue culture flasks by RPMI-1640 medium supplemented with 10% fetal

bovine serum (FBS), 100 U/mL penicillin, and 0.1 mg/mL streptomycin, pH 7.4. In a humidified condition of 5% $CO_2$ and 95% air, the cells were maintained at 37˚C for 1 hour to enable the monocytes to adhere to the plates to acquire a pure lymphocyte suspension (>98%). After isolation of blood lymphocytes, samples were mixed with Türk's solution (1:20) and the lymphocyte number was immediately measured using a Neubauer chamber under an optical microscope (Nikon, Melville, NY).

### CK and LDH determination

Following the method established by Zammit & Newsholme [10], plasma CK isoforms (NAC and MB) and LDH activities were measured using kits supplied by Bioclin Diagnostics (São Paulo, SP, Brazil). Control plasma was used to check the precision and accuracy of the assay, with a maximum error of 5% (linear correlation coefficient for CK and LDH was 0.999 and 0.9992, respectively).

### Measurement of reactive oxygen metabolites

Hydroethidine (1 μM) was added to the lymphocytes ($1.0 \times 10^6$ cells/mL) in the incubation medium, and the cells were treated immediately with phorbol myristate acetate (PMA) (54 ng/mL). The cells were incubated for 30 minutes prior to assessment of ROS production. The samples were assayed in PBS supplemented with $CaCl_2$ (1 mM), $MgCl_2$ (1.5 mM), and glucose (10 mM) at 37ºC in a final volume of 0.3 mL. Briefly, hydroethidine (a reduced derivative of ethidium bromide) is intracellularly oxidized by oxygen radicals, being converted into ethidium bromide that tightly binds to DNA and presents a strong red fluorescence. After the calibration by fluorescent beads and using the FL3 channel of a BD Accuri flow cytometer (Becton Dickinson, CA, USA), fluorescence was measured and 10,000 events were analysed per experiment [3, 11].

### Expression of cell adhesion molecules

The expression of CD62L (BD Biosciences, NJ, USA) was evaluated on the surface of lymphocytes ($1.0 \times 10^6$ cells/mL) by using a flow cytometer (Becton Dickinson, San Juan, CA, USA). Briefly, the MEL-14 monoclonal antibody is conjugated with the fluorochrome fluorescein isothiocyanate and identifies an epitope situated in the lectin domain and CD62L binds a quantity of glycosylated, fucosylated, sulfated sialylated glycoproteins including glycam-1, MAdCam-1, and CD34. Data were obtained and measured using a BD Accuri flow cytometer (Becton Dickinson, CA, USA) and were represented as the mean of the fluorescence intensity of the fluorescein isothiocyanate. A total of 10,000 events was measured per experiment.

### Statistical analysis

The data are expressed as mean ± SEM. The characterization of outliers was determined based on the criteria of Chauvenet [12], in which values lower or higher than two standard deviations in each group were removed. After assuring normal distribution of all variables (Shapiro-Wilk test), the statistical analysis consisted of parametric tests—student t-test (INStat; Graph Pad Software, San Diego, CA, USA). Pearson's correlation was used to measure linear relationships and the significance level was set at $p < 0.05$.

## Results

Fig 1 shows that the street dance class resulted in increased CK-NAC (4.87%, $p<0.05$) and CK-MB (3.36%, $p<0.05$) (Fig 1A and 1B, respectively). However, no differences in LDH

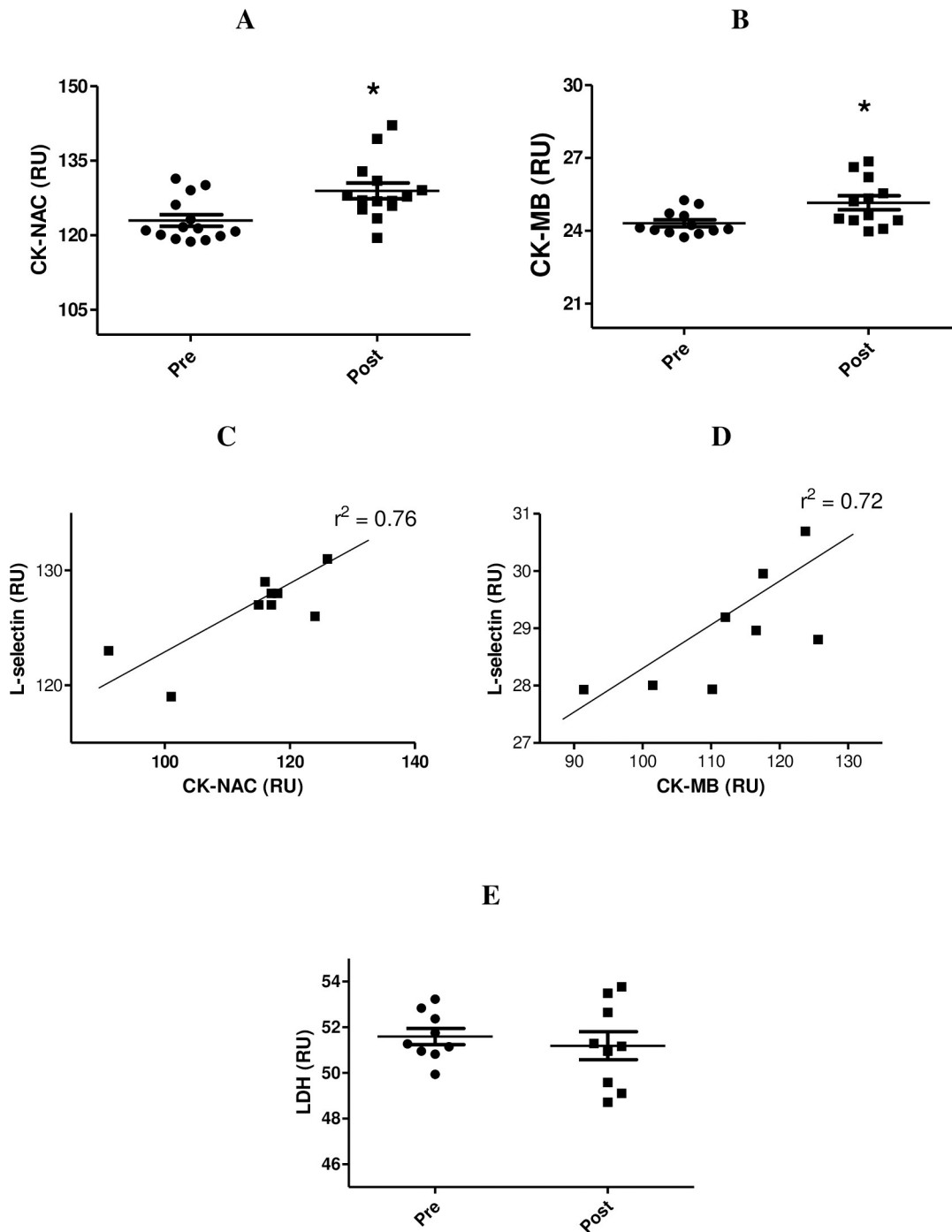

**Fig 1.** Determination of CK-NAC (A), CK-MB (B), and LDH (E) before and after the class. Data are shown as individual data points and the mean ± SEM for 9–14 participants in terms of relative unit (RU). * p< 0.05 for comparison of the values before and after the class. The correlation between CK-NAC activity and L-selectin plasma levels ($r^2$ = 0.76, p<0.05) and between CK-MB activity and L-selectin plasma levels ($r^2$ = 0.72, p<0.05) is presented in the (C) and (D), respectively.

activity was noted after the class (Fig 1E). Fig 1 also shows a positive correlation between CK-NAC activity and the plasma concentration of L-selectin ($r^2$ = 0.76, p<0.05) (Fig 1C) and between CK-MB activity and plasma concentration of L-selectin ($r^2$ = 0.72, p<0.05) (Fig 1D).

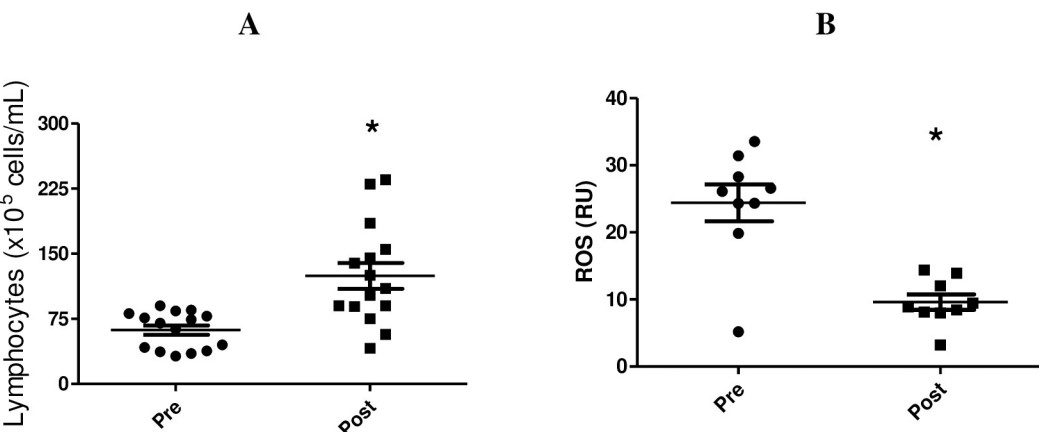

**Fig 2.** Determination of blood lymphocyte number (A) and ROS release (B) before and after the class. Data are shown as individual data points and the mean ± SEM of 9–15 participants, respectively. * p<0.0001 for comparison of the values before and after the class. ROS histogram of ten thousand events is shown as a logarithmic scale.

Lymphocytosis occurs during and immediately after physical exercise, and the magnitude is normally proportional to the exercise intensity and duration. We demonstrated that the street dance exercise increased the total lymphocyte count (2.0-fold, p<0.0001) (Fig 2A). Furthermore, after the street dance class, lymphocyte production of ROS decreased in PMA-stimulated conditions (2.5-fold, p<0.0001) (Fig 2B).

Our data demonstrated that dancing decreased lymphocyte CD62L expression (51.42%, p<0.01) (Fig 3A). There was also negative correlation between exercise intensity (% of HR reserve—averaged over full class) and CD62L expression (post-exercise) ($r^2$ = -0.73, p<0.05) (Fig 3B). Additionally, the lymphocyte gating strategy is presented in supporting information (S1).

In previous studies, our group noted an increase in plasma L-selectin after a single street dance class [2]. Herein, we showed a negative correlation between plasma concentration of L-

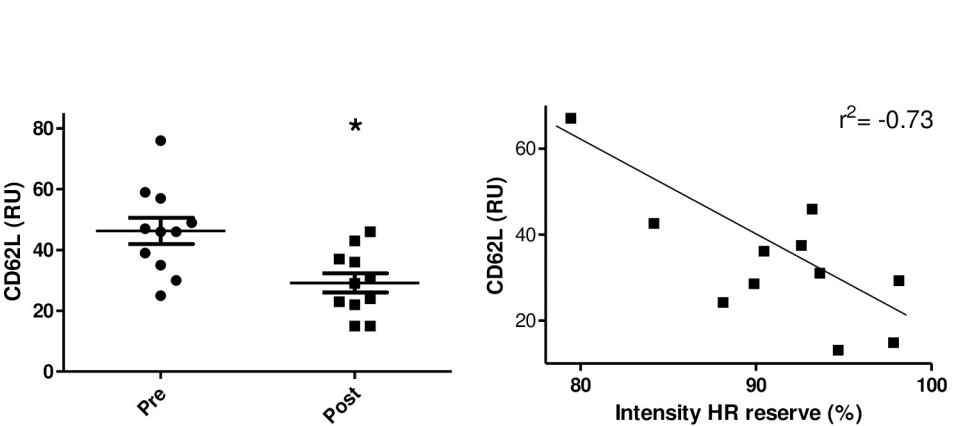

**Fig 3.** Lymphocyte CD62L expression before and after the class (A) and Pearson correlation between CD62L with the intensity of HR reserve (averaged over full class) ($r^2$ = -0.73, p<0.05) (B). Data are shown as individual data points and the mean ± SEM for 11–12 participants. * p< 0.01 for comparison of the values before and after the class. CD62L histogram of ten thousand events is shown as a logarithmic scale.

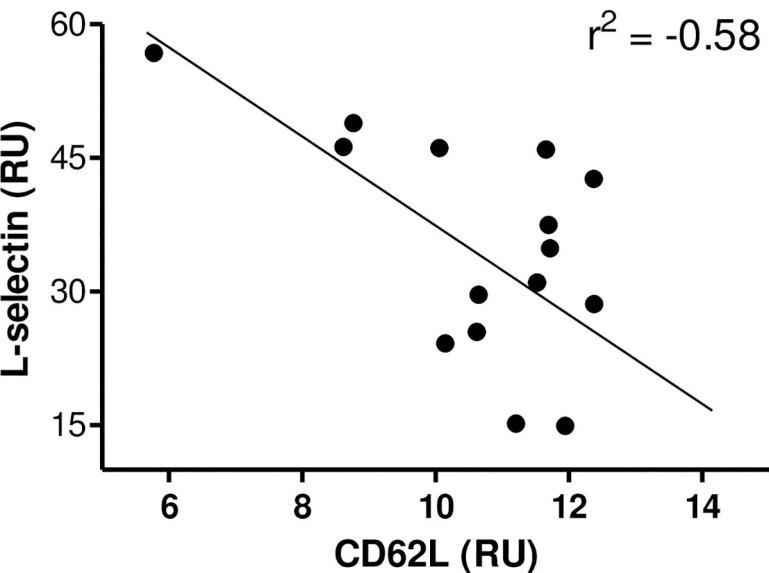

**Fig 4. Correlation between L-selectin and lymphocyte CD62L expression in street dance performers ($r^2$ = -0.58, $p < 0.05$).**

selectin (post-exercise) and lymphocyte CD62L expression (post-exercise) ($r^2$ = -0.58, $p < 0.05$) (Fig 4) in street dance performers.

## Discussion

Our data showed that a single session of street dancing was enough to increase the plasma activities of the muscle enzyme CK and increased the number of lymphocytes in circulation. In addition, our study was the first to report that lymphocyte production of ROS, after PMA stimuli, and lymphocyte CD62L expression was lower in dancers after a single class.

CD62L expression is important in leukocyte adhesion and may influence leukocyte redistribution during exercise [13]. In our study, lymphocyte CD62L expression was decreased and found at low levels after the class, and this may impair the trafficking of leukocytes in dancers. We also noted a negative correlation between the expression of CD62L in lymphocytes and exercise intensity, suggesting that the effect of street dance on the expression of adhesion molecules, a crucial event for the homing mediation of leukocytes to sites of inflammation [14], is intensity-dependent.

Recently our group identified an increase in plasma L-selectin after a single street dance class [2]. Interestingly, our data showed a negative correlation between CD62L expression from lymphocytes and plasma L-selectin. CD62L ensures that lymphocytes migrate from the blood into the lymph tissues [6] and research suggests that the reduction in CD62L+ lymphocytes after intense exercise may be due to the redistribution of CD62L- lymphocytes into the blood [15]. Moreover, Nielsen et al. noted a lower expression of CD62L after a marathon, and this reduction was followed by higher values of soluble CD62L, suggesting a shedding of adhesion molecules [16]. Since the different functions of these aspects during the recovery from exercise are not fully understood, a clinical perspective of these alterations, such as the occurrence of infections after exercise, is only hypothetical and further research is crucially needed on this matter.

The activation of lymphocytes is an important event of tissue repair. However, the inflammatory pattern must be a self-controlled event. For instance, depending on the dimension of

the spatio-temporal modulation of ROS causing oxidative stress, it can act as a secondary messenger or death stimulus [17]. After the street dance class, lymphocyte count increased but ROS production by lymphocytes was lower after PMA stimuli. Higher susceptibility to invasive microorganisms may be seen with this low production of ROS by leukocytes that come into contact with stimuli, thereby reducing the athlete's health [11, 18].

The clinical relevance of the temporary change in cell surface adhesion molecule expression on lymphocytes after intense exercise is not clear, however, it is important to remember that lymphocyte function is a significant element in the balance of inflammation and resolution of infection [19]. It is also likely that many other adhesion molecules are altered by exercise, such as very late antigen-4 (VLA-4), Mac-1 and lymphocyte function-associated antigen-1 (LFA-1) [6, 20].

The implications of this study should be interpreted in light of some limitations. First, this is a convenience sample of dancers and the sample size was small, which may limit the generalizability of the findings. Second, all participants in this study were men. Since the inflammatory response is influenced by gender [21], our data should be applied to women with caution. Also, the intensity of street dance training in future studies should investigate the HR reserve at different intervals throughout the exercise session.

In conclusion, elevated markers of muscular damage, inflammation, and decreased expression of CD62L were observed after the class, thereby suggesting potential for higher infection prevalence and impaired homing process of leukocytes following intense dance.

## Supporting information

**S1 Fig. Gating strategy of the lymphocytes isolated from human peripheral blood.** Lymphocytes of interest were gated according to forward and sideward scatter (FSC/SSC) (A). The histogram illustrates the negative control on the left (pink) and the sample with fluorescent on the right (red) (B).
(DOCX)

## Acknowledgments

The authors would like to thank the individuals who volunteered to participate in the study.

## Author Contributions

**Conceptualization:** Leandro Borges, Thaís Reis Martins.

**Data curation:** Leandro Borges.

**Formal analysis:** Leandro Borges, Renata Gorjão, Vinicius Coneglian Santos, Cesar Miguel Momesso.

**Funding acquisition:** Tania Cristina Pithon-Curi, Elaine Hatanaka.

**Investigation:** Leandro Borges, Thaís Reis Martins, Elaine Hatanaka.

**Methodology:** Leandro Borges, Renata Gorjão, Thaís Reis Martins, Vinicius Coneglian Santos, Cesar Miguel Momesso, Elaine Hatanaka.

**Project administration:** Leandro Borges.

**Supervision:** Elaine Hatanaka.

**Visualization:** Stuart R. Gray.

**Writing – original draft:** Leandro Borges.

**Writing – review & editing:** Leandro Borges, Stuart R. Gray, Tania Cristina Pithon-Curi, Elaine Hatanaka.

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
