## [Decision Letter · Decision Letter 0]

15 Jul 2020

PONE-D-20-17618

Lymphocyte activation after a high-intensity street dance class

PLOS ONE

Dear Dr. Borges,

Thank you for submitting your manuscript to PLOS ONE. After careful consideration, we feel that it has merit but does not fully meet PLOS ONE’s publication criteria as it currently stands. Therefore, we invite you to submit a revised version of the manuscript that addresses the points raised during the review process.

The reviewers have indicated the manuscript does not currently meet two of PLOS ONE's publication criteria, specifically:

3. Experiments, statistics, and other analyses are performed to a high technical standard and are described in sufficient detail.

Experiments must have been conducted rigorously, with appropriate controls and replication. Sample sizes must be large enough to produce robust results, where applicable. Methods and reagents must be described in sufficient detail for another researcher to reproduce the experiments described.

4. Conclusions are presented in an appropriate fashion and are supported by the data.

The data presented in the manuscript must support the conclusions drawn. Submissions will be rejected if the interpretation of results is unjustified or inappropriate, so authors should avoid overstating their conclusions. Authors may discuss possible implications for their results as long as these are clearly identified as hypotheses instead of conclusions.

If you feel you can address all of the reviewers comments and the manuscript will meet publication criteria #3 and #4 above, a revised manuscript can be submitted. 

We look forward to receiving your revised manuscript.

Kind regards,

Melissa M Markofski

Academic Editor

PLOS ONE

Journal Requirements:

Reviewers' comments:

Reviewer's Responses to Questions

**Comments to the Author**

1. Is the manuscript technically sound, and do the data support the conclusions?

Reviewer #1: Partly

Reviewer #2: Partly

2. Has the statistical analysis been performed appropriately and rigorously? 

Reviewer #1: I Don't Know

Reviewer #2: Yes

3. Have the authors made all data underlying the findings in their manuscript fully available?

Reviewer #1: No

Reviewer #2: No

4. Is the manuscript presented in an intelligible fashion and written in standard English?

Reviewer #1: Yes

Reviewer #2: Yes

5. Review Comments to the Author

Reviewer #1: Plos One Review Notes – Lymphocyte Activation with Dance

This study compared CK, l-selectin, lymphocyte ROS, and lymphocyte CD62L expression in blood of healthy young men before and after an hour of high intensity street dancing exercise. Compared to before-exercise, there was an increase in CK (NAC AND MB), an increase in lymphocytes, a decrease in ROS production among stimulated-lymphocytes and a decrease in CD62L+ Lymphocytes. The authors conclude that high intensity street dancing may lead to transient immunosuppression.

The authors state that all information is available in supporting documents but these were not uploaded.

Major Comments: This manuscript contributes marginally to the field of exercise immunology, as it is unique in describing immune parameters before and after high intensity street dancing. However, the authors are cautioned to not overstate their results. Most importantly, the conclusion that the bout of exercise led to transient immunosuppression should be struck from the manuscript (line 39-40, line 177-181, 196-197), as the authors have no supporting evidence that any of the dancers had suppressed immune systems. Figure 5 should also be removed, as there is no evidence provided of chronic inflammation, joint destruction, or arthritis. The data do show a lowering in lymphocyte production of ROS following generic stimulus (PMA), but without evidence of lowered function in other arenas of immune (even lymphocyte function) one cannot conclude that immune activity overall is lower or higher.

On a similar note: these results are well within a large body of literature which show a decrease in CD62L+ lymphocytes after exercise. However, most have concluded that this is due the redistribution of CD62L- lymphocytes into the blood, rather than a change in protein expression on individual cells. The results here do not show changes at a single cell level, and so again data must be interpreted with caution. The paragraph beginning on line 165 should be carefully rewritten.

Methods: Much more detail should be included in the materials and methods section. As it stands, the reader cannot fully interpret the results:

-What formula was used to estimate maximum HR? 85-89% of HRR does not seem particularly intense, particularly when considering the young adults exercised ~16 hours a week. It would be useful to show actual HR at various intervals throughout the exercise session.

-Why were only men included?

-What was the delay between cessation of exercise and the post-blood draw? If all 15 men were included on the same day (after the same bout), it seems that some may have had to wait some time before donating blood. What anti-coagulant was used?

-How were lymphocytes quantified?

-What were the inter- and intra- assay variations for CK and LDH determination?

- How was L-selectin measured?

- Line 111-118 What reaction is catalyzed that can be measured with fluorescence?

-Representative flow cytometry plot illustrating gating strategy is needed. Were all 30 samples assessed on a single day, or were they assessed on different days? If on different days, what strategies were used for calibration/standardization?

-Statistics: Paired (dependent) T-tests? Were assumptions of normality and variance met? Without this information the reader cannot understand if appropriate tests were applied.

Results:

-Where are the LDH results?

-Why is L-selectin not shown?

-How were lymphocytes counted?

Minor comments:

Abstract: define abbreviations at first use

Page 4

Line 76/77 - Both body weight and body mass listed, yet numbers are different.

Line 83 - Improper subject verb number agreement (“dancers dances”)

Line 88-91 - Description of exclusion criteria is long, yet vague. Please be more specific in regard to the rationale for the exclusion criteria.

Page 5

Line 104 - How long were cells cultured?

Line 114- Just PMA, not PMA+Ionomycin?

Page 8

Line 153 - “Our data observed” – wrong verb

Line 153 - “single session [of] street dancing” (“of” not included)

Line 155 - “Besides” – awkward word choice. “In addition” might work better

Page 9

Line 200 - The use of “dance practitioners” is awkward. Just use “dancers.”

Figure 1

Line 281 - Remove “was analyzed” from first sentence of caption

- Are axes flipped on 1d? CK-MB scale does not match 1b.

Figure 2

- Shouldn’t ROS be in fluorescence units (e.g. MFI)?

Figure 3

- CD62[L] (“L” not included in axis) units should be the same on 3a and 3b, they are not currently

Figure 4

- There are 16 data points on the graph but only 15 participants were in the study. Where did the other point come from?

Figure 5

- I suggest removal

Reviewer #2: Thank you for the opportunity to review this manuscript. The authors examined the effects of a single bout of street dancing on lymphocyte activation. They found that a single exercise session increased the number of lymphocytes found in blood, muscle enzymes CK and LDH, and decreased CD62L expression on leukocytes. Additionally, they report a negative correlation between exercise intensity and CD62L expression.

Comments:

Introduction:

- Line 55-57: “sL-selectin…preventing the binding of leukocytes to those cells” Prevents the binding to which cells?

- Can the authors define their abbreviations within the manuscript (CK, LDH, etc.).

Methods:

- Line 75: “The dancer’s volunteers”: It is unclear what the authors are referring to here

- Blood counts provided in lines 79-82: it is not clear in the manuscript if these were resting values?

- It is not clear to me how the exercise intensity was determined. For the correlations are the authors suggesting that each individual exercised at a different relative intensity? If so, this calculation needs to be better phrased.

- Measurement of reactive oxygen metabolites: Can the authors clarify what is being measured in the FL3 channel. Are the cells fluorescing and the signal is detecting in FL3? Was an antibody used?

Results/Discussion:

- Would the authors consider plotting all bar graphs in to show the raw data (individual data points) as well as the means/SEM?

- Figure 5: Systemic is misspelled

- Figure 5: Immunosuppression infection risk is not clear. It reads as if chronic inflammation added with intense dancing would lead to joint destruction/arthritis. What the authors report in this study is an acute phase of inflammation immediately after exercise, not chronic inflammation found in these individuals.

- Line 175: can the authors expand on the statement: the inflammatory pattern must be a self-controlled event?

- The link to decreased leukocyte adhesion and chronic inflammation (arthritis/arthrosis) is not clear. The authors state in their discussion that clinical consequences of their findings is not clear, yet they speculate without much evidence. Can the authors strengthen their argument for this.

6. PLOS authors have the option to publish the peer review history of their article (what does this mean?). If published, this will include your full peer review and any attached files.

Reviewer #1: No

Reviewer #2: No

---

## [Author Response · Author response to Decision Letter 0]

31 Jul 2020

COMMENTS TO REFEREES:

We thank the reviewer for all points raised. We improved the manuscript by considering your comments. All modifications, made in the manuscript, are highlighted with tracked changes.

REVIEWER #1:

MAJOR Comments.

- The authors are cautioned to not overstate their results. Most importantly, the conclusion that the bout of exercise led to transient immunosuppression should be struck from the manuscript (line 39-40, line 177-181, 196-197), as the authors have no supporting evidence that any of the dancers had suppressed immune systems.

Response: 

We removed the following statements: 

- “These data may be useful to plan strategies to protect dance practitioners against microorganism infection and chronic inflammation” (page 2, lines 40-41).

- “Since changes in cell surface adhesion molecule expression on leukocytes are commonly associated with lymphocyte homing propensity [13] and L-selectin seems to be predominantly involved in tissue-specific lymphocyte homing [15], our data suggest that a single bout of street dance class may lead in disruption to homing mechanisms and elevation in adhesion molecules” (page 9, lines 196-200).

- “our data demonstrate that immediately after a street dance class, lymphocytes from dancers were less responsive to stimuli, thereby indicating transient immunosuppression.” (page 11, lines 234-236).

- “These findings may be useful to plan strategies to protect dance practitioners against microorganism infection and chronic inflammation.” (page 11, lines 238-240).

- “Thus, these events point to the decreased efficiency of lymphocytes against infection, when exposed to pathogens, after the street dance training (Fig 5)” (page 10, lines 216-218).

- Figure 5 should also be removed, as there is no evidence provided of chronic inflammation, joint destruction, or arthritis.

Response: Figure 5 was removed.

- The results here do not show changes at a single cell level, and so again data must be interpreted with caution. The paragraph beginning on line 165 should be carefully rewritten.

Response: The authors rewrote the paragraph and briefly discussed the redistribution of CD62L- lymphocytes into the blood after exercise. We added on page 9-10, lines 200-208:

 “CD62L ensures that lymphocytes migrate from the blood into the lymph tissues [6] and research suggests that the reduction in CD62L+ lymphocytes after intense exercise may be due to the redistribution of CD62L- lymphocytes into the blood [15]. Moreover, Nielsen et al. noted a lower expression of CD62L after a marathon, and this reduction was followed by higher values of soluble CD62L, suggesting a shedding of adhesion molecules [16]. Since the different functions of these aspects during the regeneration phase are not fully understood, a clinical perspective of these alterations, such as the occurrence of infections after exercise, is only hypothetical and further research is crucially needed on this matter.”

Methods. 

-What formula was used to estimate maximum HR? 

Response: For calculation of HR maximum, we used the formula: HR maximum = 220 – age. We added this information on page 4, lines 87-88.

- It would be useful to show actual HR at various intervals throughout the exercise session.

Response: The HRR was measured at 20, 40, and 60 minutes after the class. The authors added the reviewer suggestion in the discussion section, please see page 11, lines 231-233:

“Also, the intensity of street dance training in future studies should investigate the HR reserve at different intervals throughout the exercise session.”

- 85-89% of HRR does not seem particularly intense, particularly when considering the young adults exercised ~16 hours a week.

Response: The training intensity was classified as intense according to the guidelines of the American College of Sports Medicine (Tenth edition) (information added on page 4, lines 91-93). 

-Why were only men included?

Response: the hormonal variations of the menstrual cycle modify the thermoregulation response to exercise in females, such that the core temperature during rest and exercise may differ according to the phases of the menstrual cycle (Sports Med. 2011;41(10):861-82. doi: 10.2165/11593180-000000000-00000), a fact that could influence the data in this study by increasing the variation in results and the risk of bias, thus possibly requiring a larger sample size. Additionally, we noted a greater number of male dancers during the recruitment of the participants.

-What was the delay between cessation of exercise and the post-blood draw?

Response: The blood collection delay between participants was approximately 3-4 min. Information was added in methods (Page 5, line 104).

- What anti-coagulant was used? What were the inter- and intra- assay variations for CK and LDH determination?

Response: Blood samples were drawn into BD Vacutainer® tubes containing heparin (information added on page 5, line 103). The linear correlation coefficient for CK and LDH was 0.999 and 0.9992, respectively (information was added in methods - page 6, lines 124-125).

- Line 111-118 What reaction is catalyzed that can be measured with fluorescence?

Response: hydroethidine (a decreased derivative of ethidium bromide) is intracellularly oxidized by oxygen radicals, being converted into ethidium bromide that tightly binds to DNA and presents a strong red fluorescence. Information added on page 6, lines 131-134.

-Representative flow cytometry plot illustrating gating strategy is needed. 

Response: The gating strategy was added in supporting information (please see page 8, lines: 177-178).

- Were all 30 samples assessed on a single day, or were they assessed on different days? If on different days, what strategies were used for calibration/standardization?

Response: On the day of the experiment, we calibrated the flow cytometer to confirm that the instrument was performing properly by tracking laser, detectors, and fluidic performance. This quality control was performed by fluorescent beads and according to the manufacturer's instructions (Becton Dickinson, CA, USA) (information added on page 6, line 134). All 30 samples were assessed on a single day and this information was added on page 5, lines102-103.

-Statistics: Paired (dependent) T-tests? Were assumptions of normality and variance met? Without this information the reader cannot understand if appropriate tests were applied.

Response: The authors previously performed the data normalization (Shapiro-Wilk test). Information added on page 7, lines 151-152.

Results:

-Where are the LDH results? 

Response: LDH result was added as Figure 1E e described in the results section. Please see page 8, lines 158-159. 

- Why is L-selectin not shown? How was L-selectin measured?

Response: This research is the sequence of a previous study and the plasma L-selectin value was presented in the manuscript (Title: Neutrophil Migration and Adhesion Molecule Expression after Acute High-Intensity Street Dance Exercise) published in 2018 in the journal Mediators of Inflammation. This information is stated on page 8, lines 173-174.

-How were lymphocytes counted? How were lymphocytes quantified?

Response: After isolation of blood lymphocytes (page 5, lines 101-116), samples were mixed with Türk’s solution (1:20) and the lymphocyte number was immediately measured using a Neubauer chamber under an optical microscope (Nikon, Melville, NY). The information was better described (please see page 5, lines 101-118).

MINOR comments:

Page 4

Line 76/77 - Both body weight and body mass listed, yet numbers are different.

Line 83 - Improper subject verb number agreement (“dancers dances”)

Line 88-91 - Description of exclusion criteria is long, yet vague. Please be more specific in regard to the rationale for the exclusion criteria.

Response: The authors corrected the manuscript and the change can be evidenced in line 78, line 86, and lines 95-99.

Page 5

Line 114- Just PMA, not PMA+Ionomycin?

Line 104 - How long were cells cultured?

Response: The authors used only PMA to determine ROS (incubated for 30 minutes). Information added on page 6, line 129. The cells were maintained in culture for 1 hour to enable the monocytes to adhere to the plates (information added – lines 114-115).

Abstract: define abbreviations at first use.

Line 153 - “Our data observed” – wrong verb

Line 153 - “single session [of] street dancing” (“of” not included)

Line 155 - “Besides” – awkward word choice. “In addition” might work better

Line 200 - The use of “dance practitioners” is awkward. Just use “dancers.”

Figure 1 - Are axes flipped on 1d? CK-MB scale does not match 1b.

- Figure 3: CD62[L] (“L” not included in axis) units should be the same on 3a and 3b, they are not currently.

- Figure 4: 16 data points on the graph but only 15 participants were in the study.

Figure 5: I suggest removal.

Line 281 - Remove “was analyzed” from first sentence of caption.

Response: The authors are grateful for the identified mistakes. All points were duly reviewed and corrected.

Figure 2

- Shouldn’t ROS be in fluorescence units (e.g. MFI)?

To assist visualization, the histogram of ten thousand events is shown in arbitrary units (logarithmic scale). The logarithmic scale information was added to the figure legend (lines 327-328).

REVIEWER #2:

Introduction:

- Can the authors define their abbreviations within the manuscript (CK, LDH, etc.).

- Line 55-57: “sL-selectin…preventing the binding of leukocytes to those cells” Prevents the binding to which cells?

Response: Abbreviations have been defined and the phrase was revised (please, see page 3, line 58).

Methods:

- Line 75: “The dancer’s volunteers”: It is unclear what the authors are referring to here

- Blood counts provided in lines 79-82: it is not clear in the manuscript if these were resting values?

- For the correlations are the authors suggesting that each individual exercised at a different relative intensity? If so, this calculation needs to be better phrased.

Response: we made the adjustments as suggested. Blood count corresponds to the resting values and the correlations of the study were obtained from the HR reserve of each participant. The changes can be evidenced in lines 76 and 83-84 (page 4), and lines 94-95 (page 4-5). 

- Measurement of reactive oxygen metabolites: Can the authors clarify what is being measured in the FL3 channel. Are the cells fluorescing and the signal is detecting in FL3?

Response: The hydroethidine (a decreased derivative of ethidium bromide) is intracellularly oxidized by oxygen radicals, being converted into ethidium bromide that tightly binds to DNA and presents a strong red fluorescence. This fluorescence is measured in the FL3 channel. Explanation of the method was described on page 6, lines 131-135).

Results/Discussion:

- Would the authors consider plotting all bar graphs in to show the raw data (individual data points) as well as the means/SEM?

Response: The authors plotted the data as requested by adding the graphs with raw data (individual data points) and they can be found as Supplementary Materials (Figure S1).

- Figure 5: 

- Systemic is misspelled

- Immunosuppression infection risk is not clear. It reads as if chronic inflammation added with intense dancing would lead to joint destruction/arthritis. What the authors report in this study is an acute phase of inflammation immediately after exercise, not chronic inflammation found in these individuals.

Response: Thank you for the observation. Regarding Figure 5, the authors agree with the points raised by #reviewer1 and chose to remove the figure. 

- Line 175: can the authors expand on the statement: the inflammatory pattern must be a self-controlled event?

Response: The statement was expanded and a classic example was added (please see page 10, lines 210-212). 

- The authors state in their discussion that clinical consequences of their findings is not clear, yet they speculate without much evidence. Can the authors strengthen their argument for this. 

Response: The authors agree with reviewers #1 and #2 regarding the precaution to not overstate the results. Thus, it was removed from the following statements:

- “These data may be useful to plan strategies to protect dance practitioners against microorganism infection and chronic inflammation” (page 2, lines 40-41).

- “Since changes in cell surface adhesion molecule expression on leukocytes are commonly associated with lymphocyte homing propensity [13] and L-selectin seems to be predominantly involved in tissue-specific lymphocyte homing [15], our data suggest that a single bout of street dance class may lead in disruption to homing mechanisms and elevation in adhesion molecules” (page 9, lines 196-200).

- “our data demonstrate that immediately after a street dance class, lymphocytes from dancers were less responsive to stimuli, thereby indicating transient immunosuppression.” (page 11, lines 234-236).

- “These findings may be useful to plan strategies to protect dance practitioners against microorganism infection and chronic inflammation.” (page 11, lines 238-240).

- “Thus, these events point to the decreased efficiency of lymphocytes against infection, when exposed to pathogens, after the street dance training (Fig 5)” (page 10, lines 216-218).

---

## [Decision Letter · Decision Letter 1]

4 Sep 2020

PONE-D-20-17618R1

Lymphocyte activation after a high-intensity street dance class

PLOS ONE

Dear Dr. Borges,

Thank you for submitting your manuscript to PLOS ONE. After careful consideration, we feel that it has merit but does not fully meet PLOS ONE’s publication criteria as it currently stands. Therefore, we invite you to submit a revised version of the manuscript that addresses the points raised during the review process.

Please address the reviewers' comments, especially the ones regarding the figures and confusion of a statement when the manuscript was revised. 

We look forward to receiving your revised manuscript.

Kind regards,

Melissa M Markofski

Academic Editor

PLOS ONE

Reviewers' comments:

Reviewer's Responses to Questions

**Comments to the Author**

1. If the authors have adequately addressed your comments raised in a previous round of review and you feel that this manuscript is now acceptable for publication, you may indicate that here to bypass the “Comments to the Author” section, enter your conflict of interest statement in the “Confidential to Editor” section, and submit your "Accept" recommendation.

Reviewer #1: (No Response)

Reviewer #2: All comments have been addressed

2. Is the manuscript technically sound, and do the data support the conclusions?

Reviewer #1: Yes

Reviewer #2: Yes

3. Has the statistical analysis been performed appropriately and rigorously? 

Reviewer #1: Yes

Reviewer #2: Yes

4. Have the authors made all data underlying the findings in their manuscript fully available?

Reviewer #1: Yes

Reviewer #2: Yes

5. Is the manuscript presented in an intelligible fashion and written in standard English?

Reviewer #1: Yes

Reviewer #2: Yes

6. Review Comments to the Author

Reviewer #1: Major Comments: None. It appears the authors addressed the major concern RE: overstating the results.

Medium Comments:

It is unclear if CK, LDH, and L-selectin were measured from plasma or serum. The blood collection methods indicate blood collection into heparinized tubes (suggesting plasma analysis), yet enzymatic activity methods indicate using kits for serum activity determination. Furthermore, the text of the results indicates plasma analysis but the figure captions suggest serum. Which is it?

Minor Comments: (For minor grammatical errors, a suggested insert is shown in brackets)

Line 58: CD62L is mentioned without being introduced as an alternate name for L-selectin. It should be noted at this point or introduced as such prior.

Line 64: “In the present [study], …”

Line 66/67: It is not specified in what tissue the activities of CK and LDH were measured.

Line 84: (sentence fragment) “Monitored by Polar…” Sentence needs a subject. For example, “Heart rate (HR) was monitored continuously by Polar XXX heart rate monitors during exercise.”

Line 90: “…any exercise for 72 hours before a dance class…” "a" should be replaced by "the"

Line 96: Suggest that the statement regarding batch sample analysis be moved to line 98 following “...3-4 minutes.” as its own sentence for clarity.

Line 98: “…and the blood collection delay [following exercise] between participants was approximately 3-4 minutes.”

Line 122: Rephrase “The release of ROS was incubated for 30 minutes.” Perhaps: “The cells were incubated for 30 minutes prior to assessment of ROS production.”

Line 125: "Reduced" rather than "decreased" is the appropriate chemical term in this instance.

Lines 152/153: Serum/plasma issue (see medium comments)

Line 164/169: Suggested that final sentence regarding gating strategy (line 169) be moved to line 164 to consolidate flow cytometry results.

Line 174: “lymphocytes [in circulation].”

Line 192: Suggest replacing “the regeneration phase with "recovery from exercise”

Line 208: “molecules are..."Replace "induced" with "altered”

Line 219: "In conclusion..." replace "pieces of evidence of" with ”elevated markers of"

Line 220: “…suggesting [potential for] higher…”

Line 221: “…process of leukocytes [following intense dance]."

Figures

Generally, line graphs (Fig 1, 3, 4) should include R2 values associated with the line

Figure 1: L-selectin is on different axes between 1c and 1d. This is a little confusing.

Figure 4: X-axis STILL labelled CD62 not CD62L

Reviewer #2: Thank you for responding to my comments. Only 2 minor comments:

-Bar Graphs: I don't think it is necessary to include a bar graph in the manuscript and a scatter plot with bars in the supplemental figures. For transparency, I feel that all figures should be presented as scatter plot with bars and should be included in the main manuscript- however I will let the editor decide what is best.

-Lines 210-212 are unclear and very confusing. Please consider rephrasing or dropping the added sentences.

7. PLOS authors have the option to publish the peer review history of their article (what does this mean?). If published, this will include your full peer review and any attached files.

Reviewer #1: No

Reviewer #2: No

---

## [Author Response · Author response to Decision Letter 1]

4 Sep 2020

COMMENTS TO REFEREES:

The authors would like to thank the Academic editor for identifying this study as having

merit for publication in PLOS ONE. The authors also appreciate the critical comments provided by all the reviewers. We have worked to address all the comments and incorporate all the suggestions provided by all the reviewers. All modifications, made in the manuscript, are highlighted with tracked changes.

Reviewer #1:

* MEDIUM Comments*

- “It is unclear if CK, LDH, and L-selectin were measured from plasma or serum. The blood collection methods indicate blood collection into heparinized tubes (suggesting plasma analysis), yet enzymatic activity methods indicate using kits for serum activity determination. Furthermore, the text of the results indicates plasma analysis but the figure captions suggest serum. Which is it?”

- Lines 152/153: Serum/plasma issue (see medium comments).

RESPONSE: CK, LDH, and L-selectin were measured from plasma and this information was corrected (please see page 6, line 122; page 15, lines 325-326).

*MINOR Comments*

- Line 58: CD62L is mentioned without being introduced as an alternate name for L-selectin. It should be noted at this point or introduced as such prior.

- Line 66/67: It is not specified in what tissue the activities of CK and LDH were measured.

RESPONSE: The CD62L information was added on page 3, line 56. CK and LDH were measured from plasma (page 3, line 68).

- Line 64: “In the present [study], …”

- Line 84: (sentence fragment) “Monitored by Polar…” Sentence needs a subject. For example, “Heart rate (HR) was monitored continuously by Polar XXX heart rate monitors during exercise.”

- Line 90: “…any exercise for 72 hours before a dance class…” "a" should be replaced by "the"

- Line 96: Suggest that the statement regarding batch sample analysis be moved to line 98 following “...3-4 minutes.” as its own sentence for clarity.

- Line 98: “…and the blood collection delay [following exercise] between participants was approximately 3-4 minutes.”

- Line 122: Rephrase “The release of ROS was incubated for 30 minutes.” Perhaps: “The cells were incubated for 30 minutes prior to assessment of ROS production.”

- Line 125: "Reduced" rather than "decreased" is the appropriate chemical term in this instance.

- Line 164/169: Suggested that final sentence regarding gating strategy (line 169) be moved to line 164 to consolidate flow cytometry results.

- Line 174: “lymphocytes [in circulation].”

- Line 192: Suggest replacing “the regeneration phase with "recovery from exercise”

- Line 208: “molecules are..."Replace "induced" with "altered”

- Line 219: "In conclusion..." replace "pieces of evidence of" with ”elevated markers of"

- Line 220: “…suggesting [potential for] higher…”

- Line 221: “…process of leukocytes [following intense dance]."

RESPONSE: All grammatical and relocation suggestions were considered and can be observed at: page 3 (line 66), page 4 (lines 87-88, line 94), page 5 (lines 103-104, line 104), page 6 (lines 130-131, line 133), page 8 (lines 173-174), page 9 (line 182), page 10 (line 206, line 223), page 11 (line 236), page 11 (lines 238-239).

Figures:

- Generally, line graphs (Fig 1, 3, 4) should include R2 values associated with the line

- Figure 1: L-selectin is on different axes between 1c and 1d. This is a little confusing.

- Figure 4: X-axis STILL labelled CD62 not CD62L.

RESPONSE: The authors included R2 (Fig 1, 3, 4), L-selectin from Fig1c and Fig1d was positioned on the same axis, and Fig4 was labelled as CD62L.

Reviewer #2:

Only 2 MINOR comments:

- Bar Graphs: For transparency, I feel that all figures should be presented as scatter plot with bars and should be included in the main manuscript- however I will let the editor decide what is best.

RESPONSE: As suggested, all figures included in the main manuscript are presented as scatter plot (instead of bar graphs). Please see figures 1, 2, and 3.

- Lines 210-212 are unclear and very confusing. Please consider rephrasing or dropping the added sentences.

RESPONSE: The sentence was dropped (page 10, lines 224-226).

---

## [Editor Report · Decision Letter 2]

9 Sep 2020

Lymphocyte activation after a high-intensity street dance class

PONE-D-20-17618R2

Dear Dr. Borges,

We’re pleased to inform you that your manuscript has been judged scientifically suitable for publication and will be formally accepted for publication once it meets all outstanding technical requirements.

Kind regards,

Melissa M Markofski

Academic Editor

PLOS ONE
---

## [Editor Report · Acceptance letter]

11 Sep 2020

PONE-D-20-17618R2 

Lymphocyte activation after a high-intensity street dance class 

Dear Dr. Borges:

I'm pleased to inform you that your manuscript has been deemed suitable for publication in PLOS ONE. Congratulations! Your manuscript is now with our production department. 

Kind regards, 

on behalf of

Dr. Melissa M Markofski 

Academic Editor

PLOS ONE